# Physics in the Loop: Integrating Biomechanics-Derived Training Data into a Neural Ordinary Differential Equation-Based Deformable Registration Framework

**Wensi Wu**[*1]                                                        WUW4@CHOP.EDU

**Yifan Wu**[*2]                                                        YFWU@SEAS.UPENN.EDU

**Analise M. Sulentic**[1]                                             SULENTICA@CHOP.EDU

**James C. Gee**[2]                                                     GEE@UPENN.EDU

**Alison M. Pouch**[2]                                      POUCH@PENNMEDICINE.UPENN.EDU

**Matthew A. Jolley**[1]                                               JOLLEYM@CHOP.EDU

[1] *Children's Hospital of Philadelphia, Philadelphia, PA, USA*

[2] *University of Pennsylvania, Philadelphia, PA, US.*

## Abstract

Image registration of moving heart valves has been fundamentally challenging due to large leaflet deformations occurring over a short period of time with limited temporal image resolution. In this work, we propose integrating mechanics-derived data augmentation into deep learning-based registration frameworks to enhance the accuracy of the registration of heart valve motion. A finite element (FE) analysis is employed to generate additional physically realistic image frames of heart valves, which are then integrated with a neural ordinary differential equation-based deformable registration method to facilitate the registration process. We observe that augmenting the cardiac image sequence with FE-simulated frames better preserves the dynamic anatomy of the heart valves and outperforms traditional registration-based methods alone.

**Keywords:** Biomechanics knowledge, deformable image registration, heart valve dynamics

## 1. Introduction

Congenital heart disease is a leading cause of neonatal morbidity and mortality worldwide, affecting approximately 1% of newborns annually in the United States. Advanced imaging analysis techniques, such as image registration, provide an invaluable method to assess the morphological evolution of the heart in three dimensions, which in turn can effectively guide the clinical management of congenital heart disease. Image registration of medical images has diverse applications, including the dynamic analysis of deformation fields in the brain (Nazib et al., 2018), lungs (Risser et al., 2013), and heart chambers (Zhuang et al., 2010). However, registering a dynamic sequence of heart valves presents distinct challenges, including substantial leaflet motion across cardiac frames and the suboptimal temporal resolution for capturing large deformations occurring during the transition valve leaflet opening to closure and back. As such, we propose integrating physics-based simulations into a deep-learning-based registration framework to facilitate more accurate predictions of heart valve leaflet motion.

---

[*] Contributed equally

## 2. Methods

The proposed registration framework applied to 2D-echocardiographic images (2DE) is shown in Fig. 1. First, we simulate the heart valve motion using finite element analysis of 2DE-derived leaflet models in diastole (valve open for atrioventricular valves) (Fig. 1A). The finite element (FE) simulation provides additional intermediate frames based on simulated valve leaflet motion, represented in binary masks, that were absent in the 2DE image sequences. Subsequently, we fuse the binary masks of the "hidden" leaflet motion into the original cardiac image sequences using neural ordinary differential equation-based deformable image registration (NODEO-DIR) (Wu et al., 2022), thus augmenting the "temporal resolution" of the valve motion (Fig. 1B top). To obtain an accurate binary mask of the heart valve in systole, the last frame of the augmented cardiac image sequence is registered to the actual systolic frame to correct the deformation discrepancy between the FE-simulated and the actual systole cardiac frames (Fig. 1B bottom). The biomechanics data augmentation approach effectively reduces the large diastole-systole leaflet deformation into a smaller, manageable step for registration, without requiring perfectly accurate simulated results. NODEO-DIR is a deep-learning-based registration method that leverages ordinary neural differential equations to optimize the deformation fields between image pairs (Wu et al., 2022). The NODEO-DIR architecture is shown in Fig 1C.

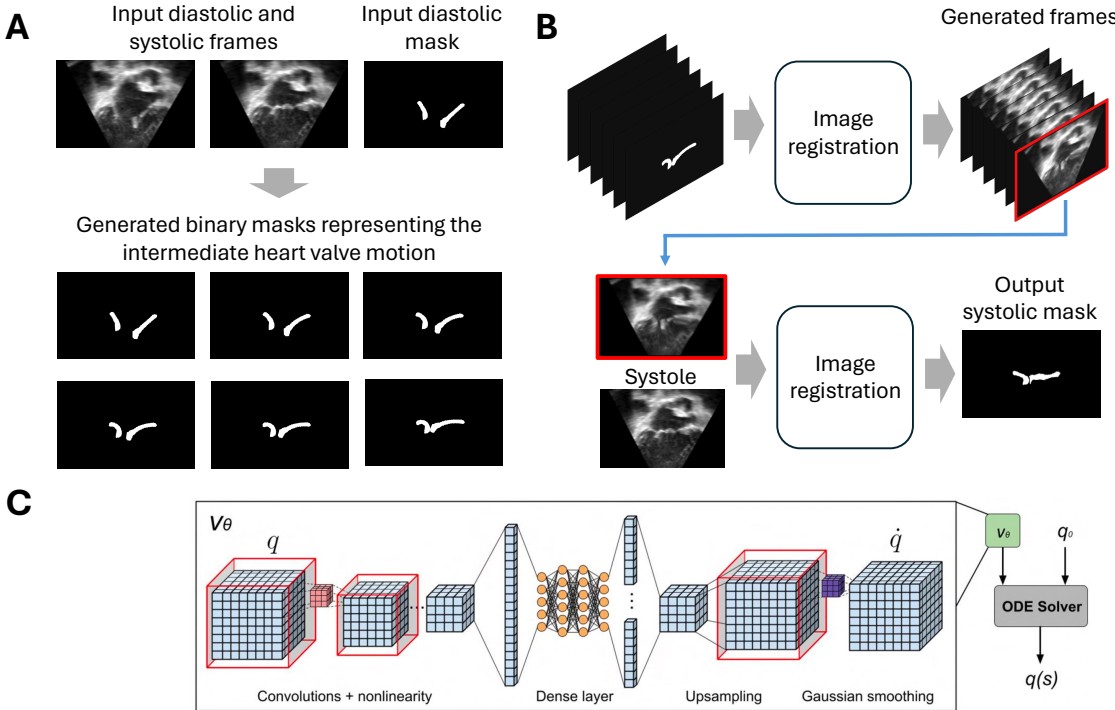

Figure 1: Overview of the proposed image registration framework augmented with biomechanics-derived image data.

## 3. Results

The preliminary results were generated from 2DE image sequences of the tricuspid valve of three children with hypoplastic left heart syndrome. We compared the registration-derived systolic binary masks with and without embedding FE-derived data augmentation prior to

the leaflet motion against manual segmentation. In general, the proposed data-augmented registration approach yielded higher Dice scores than direct registration. Further, as shown in Fig 2, integrating FE-derived image data better preserves the anatomy of the leaflets, especially near the free edge (indicated by the red arrows).

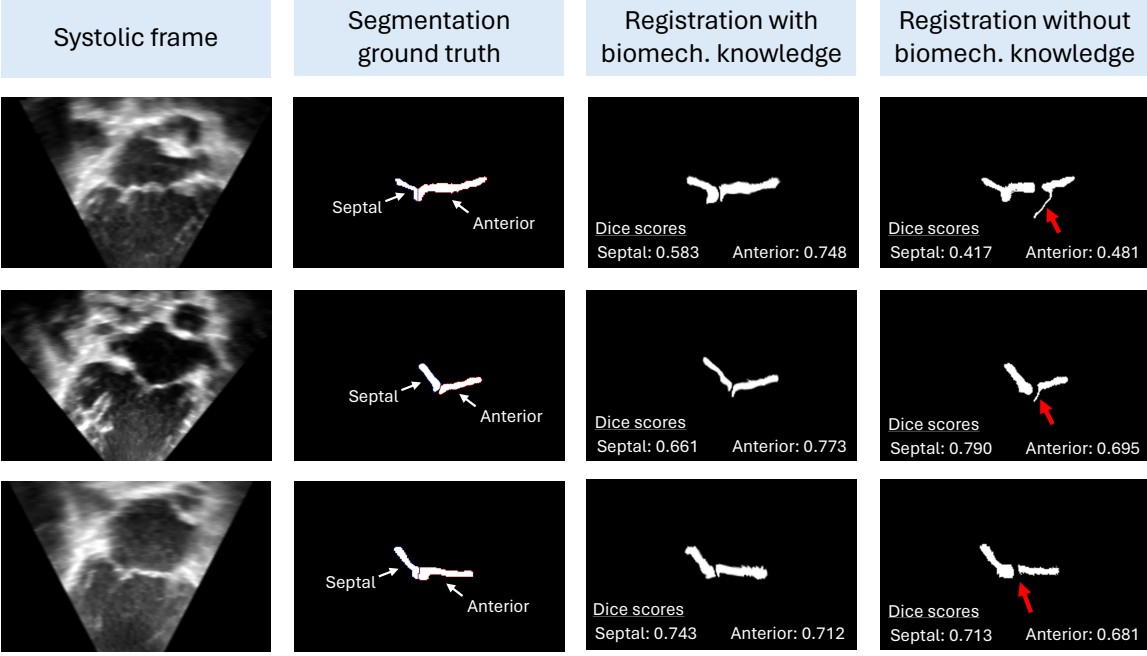

Figure 2: Comparisons of deformable image registration results with and without integration of FE-derived training data into the registration framework.

## 4. Conclusion

In this work, we present a novel approach to registering large heart valve motion leveraging synthetic images generated from physics-based simulations. Our proposed framework successfully registers the systolic phase of atrioventricular valves, accurately preserving the anatomic features of the leaflets. This reflects a major advantage of the proposed biomechanics-augmented approach, as accurately predicting the "closed" state geometry of heart valves is critical for estimating the coaptation length and evaluating valve competence. The next steps in our research include the application of the proposed framework to mitral, tricuspid, and aortic valves and translating the methods to dynamic 3D image sequences. The proposed method, once further verified, could be integrated into clinical modeling practices to facilitate understanding of valve mechanics and improve the ability to diagnose and intervene upon heart valve dysfunction. In addition, the use of physics-based simulation-derived data augmentation may be a generalizable strategy across diverse applications of machine learning.

## Acknowledgments

This work was supported by the Topolewski Pediatric Valve Center at the Children's Hospital of Philadelphia, an Additional Ventures Expansion Award, an Additional Ventures Single Ventricle Research award, NIH R01HL153166, NIH R01HL163202, NIH R01HL133889, NIH R01EB031722, and NIH K25HL168235.

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
