# OpenReview forum: "Physics in the Loop: Integrating Biomechanics-Derived Training Data into a Neural Ordinary Differential Equation-Based Deformable Registration Framework"
_MIDL.io/2024/Short_Papers — MIDL 2024 Short Papers_

### Official Review · Reviewer_hKWQ · 2024-04-23

**Confidence:** 4
**Final Rating:** 4

**Review:**

Authors propose to combine ultrasound registration and finite-element modeling (FEM) in a neural ODE framework to get a higher temporal resolution for more accurate patient-specific modeling of heart valve motion in patients with congenital heart disease. This is a technical work that combines data-driven and model-driven techniques and would be of interest to the MIDL community.

Strengths
- Interesting approach to combine FEM simulations and image registration.
- Modern registration methods based on neural ODEs.
- Results show the added value of integrating FEM information.

Weaknesses
- Paper is very dense, and not easy to follow.

---

### Decision · Program_Chairs · 2024-04-26

Accept